# Exploring the Drivers of Visitor Loyalty in the Context of Outdoor Adventure Parks: The Case of Arsenal Park in Romania

**DOI:** 10.3390/ijerph181910033

**Published:** 2021-09-24

**Authors:** Ovidiu-Ioan Moisescu, Oana-Adriana Gică, Mihaela-Corina Dorobanțu

**Affiliations:** 1Faculty of Economics and Business Administration, Babeș-Bolyai University, 400591 Cluj-Napoca, Romania; corinadorobantu23@gmail.com; 2Faculty of Business, Babeș-Bolyai University, 400038 Cluj Napoca, Romania; oana.gica@ubbcluj.ro

**Keywords:** visitor satisfaction, visitor loyalty, outdoor adventure parks, PLS-SEM, IPMA

## Abstract

Outdoor adventure parks are highly important for contemporary society, having positive social, economic, and environmental impacts. Nevertheless, to fulfill their positive role in society, and to be economically sustainable, such parks need to nurture visitor loyalty. Drawing on previous fundamental research results that ascertain that customer satisfaction has a positive influence on customer loyalty, the objective of the current research is to explore the specific elements of outdoor adventure park visitors’ satisfaction, within an applied research framework, in order to emphasize those attributes that have a significant impact on visitors’ loyalty. For that, an online survey was conducted among the visitors of Arsenal Park, Romania, one of the largest adventure parks in south-eastern Europe. Data were analyzed using partial least squares structural equation modeling (PLS-SEM). Our results show that visitors’ satisfaction with respect to the safety and equipment involved in amusement services, the ambience of the park’s food and beverage facilities, and the quality of the food are the most important satisfaction constituents for enhancing visitor loyalty in the context of outdoor adventure parks. By formatively specifying the exogeneous variables of our model (in contrast with the omnipresent reflective measurements used in previous studies), and by employing the importance–performance map analysis (IPMA), we clearly emphasize those particular aspects that are under the control of outdoor adventure parks’ managers, which significantly impact their visitors’ loyalty, as well as the way in which managers can clearly identify those attributes that need improvements.

## 1. Introduction

Outdoor adventure parks are important for sustainable development in the hospitality industry, having positive social, economic, and environmental impacts.

In relation to their social impact, by involving physical activities and social interactions in a natural environment, outdoor adventure parks positively affect their visitors’ well-being [1], life satisfaction [2], social relationships [3], as well as their health-related quality of life [4].

From an economic perspective, it can be stated that outdoor adventure parks form an important segment of the hospitality industry, serving as economic engines for the local communities in which they operate [5]. As revealed by a recent international report [6], the global leisure park market rose 6.8 percent in 2018 to an estimated USD 48.6 billion in spending, outpacing the global economic growth for the fifth consecutive year. A more recent global report [7] asserts that, despite the COVID-19 pandemic outbreak, which significantly affected the industry, the global leisure parks market is expected to quickly recover and grow at a compound annual growth rate of 11.5% to reach almost USD 89 billion by 2025.

Regarding the environmental role of outdoor adventure parks, previous studies have shown that the natural environment plays an important role in attracting adventure tourists, being a core component of their experiences [8]. Consequently, due to the economic importance of outdoor adventure parks, local communities involved in the industry are directly interested in protecting their natural environment, as it represents an essential requirement for attracting visitors. Moreover, recent research has shown that outdoor adventure parks’ visitors become more conscious and supportive of natural conservation, while owners and managers of such parks, as well as of adjacent businesses, also seek to improve their conservation and visitor education efforts [9].

In order for outdoor adventure parks to fulfill their social, economic, and environmental role in contemporary society, it is of the utmost importance that people (both residents, and from outside the community) visit such parks and get involved in active leisure activities habitually and periodically, while also stimulating their friends and acquaintances to do so. In other words, customer loyalty—in this case, park visitor loyalty—needs to be developed and nurtured. 

Over the years, both practitioners and scholars have acknowledged that increasing customer loyalty can produce several relevant economic benefits. More specifically, retaining customers and stimulating positive word of mouth have been proven to be positively associated with economic sustainability, revenues predictability, and long-term profitability [10,11,12]. Consequently, as with any organization that needs to be profitable (or at least to cover its expenses, in the not-for-profit case), outdoor adventure parks need to identify those particular aspects that are under their control, and that significantly impact their visitors’ loyalty. Further on, by improving those aspects that are deemed relevant, outdoor adventure parks can develop sustainably, via creating, maintaining, and increasing visitor loyalty.

To date, several studies have been conducted to explore the factors that drive visitor loyalty for various types of leisure parks. However, as our literature review conducted within Clarivate’s Web of Science journals shows (see Table 1), in most of these studies, the exogeneous variables included in the investigated causal models were either too abstract (e.g., emotional dimensions), or too general/broad (e.g., quality of experience), with rather limited practical implications for outdoor adventure parks, or consisted of a narrow set of specific variables, ignoring several particular attributes that could potentially impact visitor loyalty in the specific context of outdoor adventure parks.

The current study explores the drivers of outdoor adventure park visitor loyalty, adopting a practical and applied approach, by taking into account a substantial set of industry-specific attributes as potential predictors of loyalty, related to the following two indispensable components of any outdoor adventure park: amusement services and food and beverage services, respectively. By formatively specifying the exogeneous variables of our model (in contrast with the omnipresent reflective measurements used in previous studies), and by employing the importance–performance map analysis (IPMA) within partial least squares structural equation modeling (PLS-SEM), we clearly emphasize those particular aspects that are under the control of outdoor adventure parks managers, significantly impact their visitors’ loyalty, and need to be maintained/improved. Moreover, the procedure used in this research can serve as a model/tutorial that can be used by the management of any outdoor adventure park in order to enhance visitor loyalty, and, consequently, improve the economic sustainability of the park’s operations.

## 2. Theoretical Framework

### 2.1. Outdoor Adventure Parks

Outdoor adventure parks are a particular type of leisure park, in which risk, adventure, nature, and physical activity represent key features [1], a depiction that is in line with the definition of adventure tourism as provided by the Adventure Travel Trade Association [29]. Nonetheless, due to their variety, both in terms of the natural settings and leisure activities involved, defining outdoor adventure parks represents a challenging task. 

In most cases, leisure parks are a capital intensive, highly developed, user-oriented, human-modified, and recreational environment [28]. The modern leisure park industry began with Coney Island, in the late 19th century, which incorporated new technologies to provide exciting rides and audiovisual spectacles; since then, the industry has experienced almost two centuries of evolution, achieving a mature stage of development [30].

Generically stated, adventure leisure activities utilize interactions with the natural environment that contain elements of real or apparent danger, in which the outcome can be influenced by the participant and circumstances [31]. The development of adventure parks can be attributed to the so-called commodification of adventure tourism [32], a phenomenon that implied a switch from activities that were perceived as high-risk, high-difficulty, and usually reserved for those with advanced skills, to a choreographed and packaged set of more accessible leisure activities, with diluted associated risk. As a result, in an adventure park, all the activities are aimed at minimizing the objective risk, conferring a relatively safe environment, ensuring a safety–risk balance, and, at the same time, offering adrenaline-fueled experiences [1].

### 2.2. Visitor Satisfaction

Customer satisfaction, a concept that has been widely debated in the literature, and that has become a mantra in hospitality and tourism research, is traditionally associated with the difference between expectations and actual performance, with reference to a product or service. Thus, satisfaction can be defined as consumers’ response to the evaluation of the perceived discrepancy between prior expectations and actual performance, after the consumption/usage of the product/service [33]. According to this expectation–(dis)confirmation theory, satisfaction occurs when the performance confirms or exceeds the expectations. Still, satisfaction needs to be defined within a broader scope, referring to the overall evaluation of the consumption/usage experience [34]. Therefore, customer satisfaction can be depicted as a complex construct representing an overall measure of how content customers are with a product or service, considering all the potential aspects that add-up to this evaluation [35].

In the specific context of leisure parks, previous research has investigated the following various relevant particular park-related attributes that are usually involved in the consumption experience, and that eventually form the overall measure of visitor satisfaction: the park’s physical environment, with attributes such as ambiance/atmosphere, cleanliness, equipment, etc. [13,16,19,25,26,28], visitors’ interaction with staff [13,14,16,18,20,25,26,27,28], prices and perceived value for money [5,19,21,25,26,36], food quality and/or variety [5,16,19,20,26,28,36,37], crowdedness and waiting time [5,19,26,28,37], or safety and security [16,26,36,37].

Besides these directly observable and operationalizable aspects of leisure parks’ customer experience, previous research has also examined various emotional dimensions of the visitor experience such as immersion, fun, thrill, surprise, novelty, etc. [14,21,22,24,25,26]. Despite having a potential important role in the overall visitor experience, such emotional facets of the leisure park customer experience are difficult to assess and manage in an applied manner.

### 2.3. Visitor Loyalty

Simply stated, customer loyalty is a subjective behavior, resulting from complex psychological processes, which is expressed over time by consumers with respect to one or more alternative brands out of a set of brands [38]. This usually involves repeat purchase behavior and positive word of mouth. Customer loyalty refers not only to behaviors, but also to cognitive and affective dimensions, being a deeply held consumer commitment that results in rebuying one or more preferred brands consistently over time, despite various contextual influences that might cause a consumer to switch brands [39]. Equally, in the absence of a psychological commitment and a positive brand attitude, the simple repurchase of one or more brands does not mean a consumer is truly loyal [40].

It is, nowadays, generally acknowledged that satisfaction makes customers more psychologically attached to brands [41], and more prone to become loyal [42]. Hence, there is a general consensus in the literature that customer satisfaction represents an essential antecedent of customer loyalty [39]. In other words, a satisfied consumer is more likely to repurchase a brand and to recommend it to his/her friends and acquaintances.

In the specific context of leisure parks, previous studies investigating the relationship between visitor satisfaction, on one hand, and behavioral intentions to revisit and/or recommend the park, on the other hand, have come to a similar conclusion: there is a strong positive influence of visitor satisfaction on visitor loyalty (e.g., [13,15,17]). However, as previously stated, visitor satisfaction represents a complex construct [35], which encompasses a substantial set of aspects that add-up to visitors’ evaluations of their experience. For instance, a visitor might be very satisfied with the staff and the equipment of the park’s amusement amenities, but, at the same time, unsatisfied with the staff and the food at the parks’ restaurant facilities. In order to produce practical implications, research on the impact of visitor satisfaction on visitor loyalty should take into account not only the overall relationship between satisfaction and loyalty. More specifically, we need to study the particular impacts of visitors’ satisfaction with various establishments of the park (e.g., amusement, restaurant, etc.), as well as the particular influences of visitors’ satisfaction with each constituent of each amenity (e.g., satisfaction with the safety of the amusement services, satisfaction with the quality of the food provided by the park’s restaurant facilities, etc.).

Considering the current’s research focus—outdoor adventure parks—and the two indispensable components of any such leisure facility—amusement services and food and beverage facilities, respectively—we issued the following research hypotheses:

**Hypothesis** **1** **(H1):**
*Visitors’ satisfaction with the amusement services offered by an outdoor adventure park positively predicts visitors’ loyalty.*


**Hypothesis** **2** **(H2):**
*Visitors’ satisfaction with the restaurant services offered by an outdoor adventure park positively predicts visitors’ loyalty.*


## 3. Research Design and Data Collection

The current research is aimed at identifying the most relevant constituents of visitor satisfaction for enhancing visitor loyalty, in the context of outdoor adventure parks. Drawing on previous fundamental research results that ascertain that customer satisfaction has a positive influence on customer loyalty, our goal is to explore the specific elements of outdoor adventure park visitors’ satisfaction, within an applied research framework, and to emphasize those attributes that have a significant impact on visitors’ loyalty.

In order to accomplish our research goal, we focused on Arsenal Park, one of the largest adventure parks in south-eastern Europe. Arsenal Park was launched in 2009 after an investment of 25 million euros. Being built on the ruins of a former military base and munitions factory, the park has a military theme, occupying 88 hectares of land in the proximity of Orăștie, a city in the Romanian county of Hunedoara. Arsenal Park offers its visitors various amusement services related to adventure activities such as zip lines, power-fan jumps, armored rides, paintball, firearms shooting, etc., as well as a large military themed restaurant with mostly Romanian cuisine [43]. The geographical position of the park, as well as a map depicting the activities and services provided to its visitors can be seen in, respectively. Visitors can get to Arsenal Park from any part of Romania by train, by bus, or by car. However, as the park is located within a 10-minute drive to a highway exit, most visitors come here by car. As for prices, Arsenal Park charges slightly above the average tariffs for its services, compared to other outdoor adventure parks in Romania. However, it does not specifically target high income visitors, its prices being affordable to any average income family in Romania.

Furthermore, an online survey was conducted among the park’s recent visitors, targeted at assessing their satisfaction with and loyalty to Arsenal Park. The invitation to participate in the study alongside the link to the online questionnaire was disseminated via social media, in 2019, on the park’s Facebook page (facebook.com/ArsenalPark), as well as on various tourism/travel-related Facebook pages or groups. The invitation and questionnaire were only addressed to Romanian visitors, being available exclusively in Romanian. However, this did not significantly impact the representativeness of the study, as, according to the management of the park, the vast majority of the parks’ visitors come from Romania, of which more than half are from Hunedoara, the county where the park is located.

More than 200 park visitors filled in the questionnaire. Only those who had experienced both the amusement and restaurant services offered by Arsenal Park were kept in the sample (missing answers for amusement or restaurant services, respectively, were treated with case-wise deletion). Thus, the final validated sample included a total of 147 visitors of Arsenal Park (see Table 2 for sample demographics). As it can be seen, most respondents reside in the county of Hunedoara, which resembles the investigated population’s overall residence-based structure.

In order to draw-up an appropriate scale for measuring Arsenal Park’s visitors’ satisfaction, potentially relevant attributes of visitor experience were identified, as a first step, by qualitatively analyzing reviews posted by the park’s visitors on Google (more than 2000 reviews posted up to 2019) and TripAdvisor (about 80 reviews posted up to 2019). Only reviews including specific textual feedback were scrutinized, and, of these, only those reviews that specifically pointed out negative or positive aspects related to outdoor activities, or food and beverage were taken into account. Both positively and negatively described aspects of visitors’ experience were considered in order to draw-up the initial list of potentially relevant attributes from a satisfaction measurement perspective.

In a second step, the drawn-up list was filtered by retaining only those attributes that were emphasized as potential satisfaction determinants in previous studies regarding leisure parks, as our literature review had shown (see Table 1). For example, the level of noise produced by groups of visitors, even though mentioned in some reviews, was not kept in the attributes’ list, as it had not been previously considered in the literature as a relevant constituent of leisure park satisfaction. Thus, we made sure that the items included in our measurement scale were relevant considering both the theoretical perspective, and the particular context of Arsenal Park.

Eventually, a shortlist comprising the most relevant elements was drawn up for measuring visitors’ satisfaction with the parks’ amusement services and its restaurant: cleanliness, equipment, prices, safety, staff, and waiting time, for the amusement services; ambience, food quality, menu, prices, staff, and waiting time, for the restaurant. All these elements were further included in the online questionnaire as satisfaction items with answering options ranging from one = very unsatisfied to seven = very satisfied. Consequently, visitors’ satisfaction was measured via two formative latent variables, each item/indicator capturing a specific constituent of satisfaction. Additionally, two single item questions were included in the questionnaire to reflect overall satisfaction with the park’s amusement and restaurant services, respectively, using similar answering options. These two additional questions were specifically included in the questionnaire so that we could afterward validate the two formative latent variables depicting visitor satisfaction. 

Visitor loyalty was measured reflectively using a scale with four items, each item with Likert answering options ranging from one = strongly disagree to seven = strongly agree, adapted from Ma et al. [24] and Slåtten et al. [27], referring to their revisit and recommendation intentions: I intend to revisit the park in the future (LOY1), I will say positive things about the park (LOY2), I would revisit the park, if I were in the area (LOY3), I will recommend the park to others (LOY4).

A detailed view of the items included in the questionnaire in order to measure visitors’ satisfaction and loyalty can be seen in Appendix C.

Considering the exploratory and prediction-oriented nature of our research, a parsimonious path model for the impact of visitor satisfaction on visitor loyalty was proposed for further analysis using PLS-SEM (Figure 1).

PLS-SEM was employed for data analysis for two reasons. Firstly, PLS-SEM integrates and handles reflective and formative latent variables within the same path model [44], our proposed model incorporating both formative (visitors’ satisfaction towards amusement services and restaurant, respectively) and reflective measurements (visitor loyalty). Secondly, PLS-SEM is suited for exploring relationships and is focused on prediction, estimating model parameters so that the explained variance of the dependent variables is being maximized [44]. Considering the fact that our research is exploratory and prediction-oriented, with the goal to explore the specific elements of park visitors’ satisfaction and to emphasize those that have the highest impact on visitors’ loyalty, PLS-SEM represents the best option in this case. Additionally, PLS-SEM works well with small sample sizes, such as in our case [44]. The software of choice for this research was SmartPLS 3 (SmartPLS GmbH, Boenningstedt, Germany) [45].

## 4. Results

### 4.1. Measurements Assessment

Following the guidelines provided by Hair et al. [44], the measurements’ assessment was conducted separately for reflective and formative constructs, using distinct procedures for each category of latent variables.

#### 4.1.1. Reflective Measurements Assessment

We firstly evaluated the construct of visitor loyalty, which was measured reflectively. Outer loadings produced by the PLS algorithm for all four loyalty items were above the threshold of 0.7 (with values between 0.903 and 0.973), suggesting high indicator reliability. Additionally, the loyalty construct exhibited very good internal consistency, with both Cronbach’s alfa (0.971) and composite reliability (0.974) values above the recommended threshold of 0.7. With respect to convergent validity, the average variance extracted value for the construct of loyalty (0.902) was above the cutoff point of 0.5, showing that the construct was convergent.

#### 4.1.2. Formative Measurements Assessment

Further on, we assessed the two formative constructs designed to measure visitor satisfaction. Firstly, in order to ensure that collinearity between the formative indicators was not an issue, we computed the variance inflation factor (VIF) for each indicator (see Table 3). As all the VIF values were below the threshold of five, we concluded that, in the context of PLS-SEM, no collinearity issues would impact our further estimations.

The next step in validating our formative measurements consisted of assessing their convergent validity. In the PLS-SEM context, the convergent validity of a formative variable reflects the extent to which it correlates positively with other measures of the same construct using different indicators. For this, researchers can conduct a redundancy analysis, testing whether the formatively measured construct is highly correlated with a reflective measure of the same construct. Using the single item questions included in the questionnaire to reflect overall satisfaction with the park’s amusement and restaurant services, we were able to run redundancy analyses for both our formative constructs. As can be seen in Figure 2, the strength of the path coefficient linking each pair of constructs has a magnitude above the cutoff point of 0.7, with R^2^ values above 0.5. Hence, we concluded that the two formative constructs exhibit convergent validity.

The next and final step of validating our formative constructs consisted of assessing the significance and relevance of their corresponding formative indicators. In order to do that, we employed the PLS-SEM bootstrapping procedure with 5000 subsamples; the results are shown in Figure 3. As can be seen, several *p* values go beyond the cutoff point of 0.05, thus revealing the fact that the corresponding formative indicators’ weights are statistically nonsignificant. However, a nonsignificant weight is not indicative of poor measurement model quality, as long as the indicator’s absolute contribution to the construct, given by its outer loading, is high (i.e., above 0.50). In our case, all the outer loadings produced using the PLS-SEM algorithm, for all the formative indicators, were above the threshold of 0.5 (outer loading values ranging from 0.656 to 0.931). Therefore, we retained all the formative indicators in our model, and fully validated the two formative constructs as proposed.

### 4.2. Structural Model Assessment

#### 4.2.1. Collinearity Checks

Before assessing the structural model, we checked for collinearity issues among the two predictor constructs. The variance inflation factor (VIF) value between amusement services satisfaction and restaurant satisfaction, as predictors visitor loyalty, was 2.65, clearly below the threshold of 5 [44], indicating that collinearity among the predictor constructs was not an issue in our structural model.

#### 4.2.2. Predictive Relevance Assessment

When using PLS-SEM, meaningful practical implications can be considered valid only if the predictive relevance of the examined model is established [46]. As the investigated relationships were based on prediction, and the goal of the research was not only to fill in a literature gap, but also to provide practical implications for the management of Arsenal Park, in particular, and for outdoor adventure parks, in general, we further assessed our model’s predictive relevance.

Firstly, we evaluated in-sample predictive relevance (or explanatory power), by examining the coefficient of determination (R^2^) for our target variable, with the results showing that a large proportion of visitor loyalty’s variance (76.9%) is explained by the two predictor constructs depicting visitor satisfaction.

Secondly, we estimated the model’s out-of-sample predictive power using the more advanced and accurate PLSPredict procedure, a holdout sample-based procedure that generates case-level predictions at an endogenous item level [47]. We used 10 folds and the same number of replications, comparing the RMSEs (root mean squared errors) of the PLS-SEM model prediction with those generated using a naïve linear model benchmark. The results are outlined in Table 4. Complying with the guidelines suggested by Shmueli et al. [47], we firstly analyzed the Q^2^_predict values of the PLS-SEM model. As all the indicators yield strictly positive Q^2^_predict values, it can be stated that it is feasible to compare the RMSEs of prediction from PLS-SEM with those from the naïve linear model benchmark. Further on, as the RMSE values yielded by the PLS-SEM prediction are lower than those generated by the naïve linear model benchmark for three out of the four indicators of visitor loyalty, we can assert that our model has a borderline high predictive power.

These results provide clear support for our model’s predictive relevance regarding outdoor adventure park visitor loyalty. Ergo, the practical implications of our results can be considered meaningful and scientifically sound.

#### 4.2.3. Structural Model Relationships Assessment

To test the hypothesized relationships, we used the PLS-SEM bootstrapping procedure with 5000 subsamples. The results, summarized in Figure 3, confirm our research hypotheses. Considering the path coefficients between the exogenous and endogenous constructs (0.482 and 0.445, with *p* < 0.001 in both cases), it can be stated that both satisfaction with amusement services (H1) and satisfaction with restaurant services (H2) have a positive and statistically significant impact on visitor loyalty, the impact of the two predictors being relatively equal.

Additionally, analyzing the formative indicator’s weights and their corresponding *p* values, we can outline those particular attributes of the amusement and restaurant services that are truly relevant for visitor loyalty. In the case of amusement services, the most influential satisfaction constituents are safety (weight = 0.392, *p* = 0.002) and equipment (weight = 0.263, *p* = 0.013), while in the case of restaurant services, ambience (weight = 0.519, *p* = 0.000) and food quality (weight = 0.408, *p* = 0.001) represent the most relevant satisfaction components for enhancing visitor loyalty. 

Visitors’ satisfaction, with the other attributes included in our construct, exhibits statistically unsignificant weights (*p* > 0.05). However, in the case of amusement services, visitors’ satisfaction with the staff and with the facilities’ cleanliness exhibit rather substantial weights (0.198 and 0.191, respectively), which, despite being unsignificant, suggest that these attributes might also play a role in boosting visitor loyalty, even though they are less important than visitors’ satisfaction with the amusement services’ safety and equipment.

Considering that heterogeneity, given by subgroups of data entailing substantially different model estimates, might produce misleading results when estimating the model based on the entire data set [48], we ran the finite mixture (FIMIX-PLS) algorithm to make sure heterogeneity was not an issue. Following Hair et al. [49], we initiated the procedure for a one-segment solution, and then rerun it for up to four segments, having in mind that for each information criteria, the optimal solution is the number of segments with the lowest value (see Table 5).

As all the information criteria (AIC3, AIC4, BIC, CAIC) point to a single-segment solution, we can conclude that heterogeneity is not an issue and does not affect our data. Therefore, the subsequent model estimates are robust [50], and the relationships and weights would not differ significantly between potential subgroups (males vs. females, day trip visitors vs. overnight visitors, or any other grouping variable).

#### 4.2.4. Importance-Performance Map Analysis (IPMA)

The IPMA [50,51] represents an extension of the standard PLS-SEM results reporting, which takes into account the average values of the latent variable scores. More specifically, the IPMA contrasts the total effects, representing the predecessor constructs’ importance in predicting a specific target construct, with their average latent variable scores indicating their performance. When computing these average values, the IPMA rescales indicator scores (initially ranging from one to seven in our study) on a range between 0 and 100, with 0 representing the lowest and 100 representing the highest performance. The overall purpose of the IPMA is to emphasize those variables and/or indicators that have a relatively high importance (i.e., total effect) for predicting the target construct, but do not exhibit a good enough performance (i.e., average latent variable scores); therefore, practical recommendations for improvements can be outlined.

Considering the parsimonious nature of our model, as well as the formative measurements of visitor satisfaction, we conducted the IPMA at indicator level, for the two satisfaction-related constructs, setting visitor loyalty as the target variable. The results are shown in Figure 4.

Based on the IPMA results, we can emphasize that visitors’ satisfaction with respect to the safety provided by the adventure parks’ amusement activities, along with the ambience of the park’s restaurant, are the most important satisfaction constituents for enhancing visitor loyalty. Additionally, considering Arsenal Park’s performance for these two attributes, it can be stated that the management of the park is doing a good job, with the scores for both these attributes being above the average. However, in relation to the ambiance provided by Arsenal Park’s restaurant to its customers, there is substantial room for improvement.

Visitors’ satisfaction with the food quality offered by the park’s restaurant, as well as with the equipment used for amusement activities, represent the next two satisfaction constituents that need to be carefully managed in order to boost visitor loyalty. As for Arsenal Park’s performance for these two attributes, even though both exhibit scores above the average, the management should consider substantial improvements with respect to the food quality offered by the park’s restaurant, in order to improve visitor loyalty.

Visitors’ satisfaction with the cleanliness of the amusement services unit of the park, and with the staff involved in providing these services, represent other potentially important facets of satisfaction that need to be carefully managed in order to maintain or nurture visitor loyalty. Even though these two indicators were not found to have statistically significant weights after the bootstrapping procedure, their importance is substantially higher compared to the other remaining indicators. In relation to Arsenal Park’s performance for these two attributes, considering their relatively good scores, as well as their lower importance, the managerial recommendation would not involve investing in a significant improvement, but would imply maintaining their performance level.

Surprisingly, visitors’ satisfaction with waiting time and prices, as well as with the restaurant’s staff or its food and beverage variety, represent unimportant facets of satisfaction, their improvement being technically unnecessary for increasing visitor loyalty. Consequently, despite Arsenal Park’s mediocre performance for visitors’ satisfaction with prices, as well as with the restaurant’s waiting time and menu, the management should not prioritize these aspects for future improvement intended at maximizing visitor loyalty.

## 5. Discussion

Our results have revealed that, despite an unanimously acknowledged strong, significant, and positive causal relationship between satisfaction and loyalty, the distinct visitor satisfaction constituents have different outcomes when it comes to visitor loyalty in the case of outdoor adventure parks.

Thus, according to our findings, the most impactful satisfaction ingredients in the case of such leisure facilities are represented by the perceived safety of the amusement services, and the ambience of the food and beverage facilities. Regarding safety, our findings are in line with those of Milman et al. [36], who concluded that safety and security were among the most important perceived attributes of leisure parks by their visitors. Similarly, Ryan et al. [26] found safety at the highest levels of importance attributed by park visitors, emphasizing the need to continuously monitor and maintain this facet of a park’s operation. The relevance of safety was also pointed out in previous studies conducted by Chen et al. [16] and Tsang et al. [37]. As compared to other categories of services in the hospitality industry, safety/security in an outdoor adventure park is one of the most important success factors, being a basic requirement, with visitors expecting leisure adventures and park experiences to have a calculable, predictable, and preferably, low risk.

In relation to the ambience of the food and beverage facilities, our finding is rather novel, with previous studies being generally focused on the overall ambiance or atmosphere of parks, not being specific with respect to certain park units, areas, or services. For instance, Basarangil [14], as well as Dong and Siu [18], included the park’s atmosphere in a “serviscape” construct, which they found as having a significant positive impact on revisit and recommendation intentions. Wu et al. [28] also concluded that a park’s general ambience positively influences revisit intention, indirectly, via overall experiential quality. Similarly, Geissler and Rucks [19] found park atmosphere to significantly influence the decision of whether to visit the park again in the future. Additionally, Torres et al. [5] outlined that well-designed physical facilities cause positive sensory experiences, and generate positive word-of-mouth. Our finding, however, specifically emphasizes the importance of the ambiance in food and beverage facilities, which are indispensable in any leisure park. This might be due to the fact that in outdoor adventure parks, people usually spend an extended amount of time and practice a variety of physical activities; therefore, they need food and beverages, and it is important for visitors to serve them in a nice, relaxing ambiance that will contribute to the overall experience.

The current study’s findings also revealed the following two other relevant constituents of visitor satisfaction that have a significant impact on visitor loyalty: the food quality offered by the park’s restaurant facilities, and the equipment used to provide amusement services. With respect to food quality, our results are in line with previous findings outlined by Geissler and Rucks [19], who point out that a park’s food quality is highly important to visitors, food and beverage services being clearly an integral part of the park experience (park restaurants often becoming attractions themselves). Similarly, Torres et al. [5] point out that the food and beverage services influence visitors’ sensory experiences and can generate word-of-mouth. Moreover, in their study, Milman et al. [36] found that the quality of food was rated above the average on importance by park’s visitors. Additionally, other studies [20,28] have investigated the impact of food quality on leisure parks’ visitor revisit or recommendation intentions and revealed significant positive indirect impacts. However, in these latter cases, the food attribute was integrated into broader constructs, with its specific influence being unclear.

Regarding visitors’ satisfaction with the equipment used to provide amusement services, which, in our research, was shown to have a significant influence on visitor loyalty, our results are compatible with those obtained by Wu et al. [28], who found that a park’s equipment can positively influence revisit intention, indirectly, via overall experiential quality. Nevertheless, equipment, as a constituent of visitor satisfaction, has been rather ignored in previous studies investigating the drivers of visitor loyalty. Our findings emphasize the importance of having modern, up to date, easy to use, secure, and full equipment for all the amusement services provided to visitors in outdoor adventure parks.

Surprisingly, our results revealed that visitors’ satisfaction with the staff, prices, or waiting time has no significant impact on visitor loyalty, regardless of whether the amusement or restaurant services are taken into account. Regarding the staff, our findings are contrary to those outlined by previous research. For instance, according to Milman et al. [36], park visitors rated the importance of having friendly and courteous staff above average. Similarly, Ali et al. [13], Jensen [20], Basarangil [14], Dong and Siu [18], Slåtten et al. [27], and Wu et al. [28] have suggested that visitors’ interaction with the park’s staff indirectly influences their intention to revisit and/or recommend a park. However, these studies have not dealt with the particular case of outdoor adventure parks. This type of park is visited for the opportunity to engage in a variety of physical activities performed individually or in groups; therefore, the interaction with parks’ staff does not significantly impact the quality of these kinds of experiences.

In relation to the prices charged within leisure parks, previous studies’ results have been inconsistent. For instance, Geissler and Rucks [19] found that visitors’ satisfaction with the total cost of the experience was the primary predictor of whether they would visit the park again. Similarly, Milman et al. [25] suggested that perceived value for money was influential in explaining the likelihood to revisit. However, Jin et al. [21] found no significant impact of visitors’ perceptions of fairly priced fees on revisit or recommendation intentions. The current study’s findings support the idea of visitor loyalty inelasticity with respect to prices, both when it comes to the core activities of an outdoor adventure park (amusement services), or to its food and beverage services. This can be explained by the fact that visiting an outdoor adventure park is rather a rare occasion so people do not consider the price when taking the decision to visit such a facility but are looking for the experience they will have.

Lastly, considering visitors’ satisfaction with the waiting time, our results are inconsistent with those provided by Geissler and Rucks [19], and Wu et al. [28], who suggest that waiting time positively influence visitors’ decision to visit a park again in the future. Nevertheless, these studies have not been focused on outdoor adventure parks, in which case visitors have lower expectations regarding this aspect and are generally more willing to accept longer waiting times, especially considering the particularities of certain adventure activities (e.g., zip lines, power-fan jumps, etc.), and the fact that they visit such parks for leisure and relaxation with family and/or friends outdoors; therefore, they are not bothered if they have to wait some time for certain activities.

Overall, our study partially confirms previous results obtained by researching the impact of visitors’ satisfaction and loyalty in the case of leisure parks. However, as this is, to the best of our knowledge, the first study to investigate the relationship in the case of outdoor adventure parks, in an applied manner, it emphasizes several satisfaction elements that have a distinct path of relationship with visitor loyalty, as compared to other types of leisure parks. These contradicting findings may be the result of the various particularities of such parks. Thus, one of the major differences between outdoor adventure parks when compared to other types of leisure parks is that they are basically much less crowded and more personal. Additionally, even though all types of leisure parks conduct regular inspections to keep their amusement infrastructure and activities safe, outdoor adventure parks need to use some of the most advanced safety equipment, as they need to provide a calculated level of risk and adventure. Additionally, while common leisure parks offer their visitors a mostly anthropic landscape, with acres of asphalt, and various human-made facilities, outdoor adventure provides visitors an environment that is as natural as possible, with as few as possible elements of built infrastructure.

## 6. Practical Implications

From a practical standpoint, our findings suggest that in order to increase visitor loyalty, outdoor adventure parks managers need to prioritize and carefully target their efforts towards improving their visitors’ satisfaction with respect to certain specific aspects of their operations, while maintaining a decent level of satisfaction with other attributes.

Thus, with respect to the core operations of an outdoor adventure park, the focus should be on improving customer satisfaction with the perceived safety provided by the parks’ amusement services, as well as with the equipment involved. Safety represents an essential requirement in outdoor adventure parks; visitors expect leisure adventure and park experiences to have a calculable, predictable, and preferably, low risk. Additionally, having modern, up to date, and easy to use equipment for all the amusement services provided to visitors is of the utmost importance.

Regarding the food and beverage services offered by an outdoor adventure park, as an integral part of the park experience, and often an attraction themselves, our findings emphasize the fact that to enhance visitor loyalty, park managers need to focus on visitors’ satisfaction with the ambiance and the food quality offered within their restaurant facilities.

As for the remaining satisfaction constituents, such as those related to staff, waiting time, or prices, for amusement and restaurant services alike, they should be treated as secondary targets within outdoor adventure parks’ efforts to enhance visitor loyalty. However, despite not having a significant impact on revisit or recommendation intentions, these attributes should be monitored and maintained at a decent level of satisfaction, because an extremely low performance would pose the risk of ruining the overall park experience, and, eventually, losing repeat visitors.

Other relevant practical implications result from our IPMA, which points out several satisfaction indicators that have a relatively high importance for predicting visitor loyalty, but do not exhibit a good enough performance in the case of the outdoor adventure park investigated within the current research—Arsenal Park. In order to increase visitor loyalty, Arsenal Park’s management should focus on improving the food quality and the ambiance provided by its restaurant, in which case, our results emphasized that there was room for substantial improvement. As for the other satisfaction constituents, the managerial recommendation for Arsenal Park would not involve investing in a significant improvement but would imply maintaining their performance level.

Of course, the latter practical recommendations are specifically addressed to Arsenal Park and cannot be generalized to other players from the industry. Nevertheless, with respect to the IPMA, our general recommendation for outdoor adventure park managers, as well as for any leisure park manager, is to periodically collect visitor satisfaction and loyalty data, to perform IPMAs, and to prioritize future improvements appropriately, based on IPMA results, so that visitor loyalty can be enhanced. For this purpose, the procedure used in the current study can serve as a tutorial that can be used by the management of any outdoor adventure park.

As a final remark, we need to point out that visitors’ loyalty cannot be nurtured while ignoring the online environment. The Internet has become an important communication media for tourism organizations, in which online travel communities significantly impact tourists’ choices [52]. In such virtual communities, travel is not so much talked about in information or functional terms, but in images, videos, and sensations that are shared [53]. Therefore, another recommendation for adventure park managers is to develop such virtual communities where their visitors can share their experiences with other people, as a manifestation of their attitudinal loyalty.

## 7. Conclusions

To fulfill their social, economic, and environmental role in the contemporary society, outdoor adventure parks need to constantly monitor and nurture their visitors’ loyalty, and to generate repeat-visit behavior and positive word-of-mouth. The current study emphasizes those particular attributes of outdoor adventure parks that can be managed and controlled, and that significantly impact visitor loyalty. By improving these aspects, outdoor adventure parks can effectively maintain and increase visitor loyalty, thus developing in an economically sustainable manner.

Our results show that visitors’ satisfaction with respect to the safety provided by the adventure parks’ amusement services, along with the ambience of the park’s food and beverage facilities, are the most important satisfaction constituents for enhancing visitor loyalty toward outdoor adventure parks. Visitors’ satisfaction with the food quality offered by park’s restaurant amenities, as well as with the equipment used to provide customers the amusement services, represent another two satisfaction elements that can significantly boost visitor loyalty. Satisfaction with other outdoor adventure park aspects such as waiting time, prices, staff, or cleanliness were not found to be statistically significant in influencing revisiting or recommendation intentions. However, despite being statistically unsignificant, satisfaction with the cleanliness of the amusement services unit of the park and with the staff involved in providing these services, represent potentially relevant aspects for nurturing visitor loyalty, their importance being substantially higher compared to the other remaining attributes taken into account.

By adopting a practical and applied approach, more specifically by taking into account a substantial set of industry-specific attributes as potential predictors of loyalty, the current research enriches the existing knowledge regarding the factors that drive outdoor adventure parks’ visitors’ loyalty, while, at the same time, provides meaningful and scientifically sound practical implications for park managers. By formatively specifying the exogeneous variables of our model (in contrast with the omnipresent reflective measurements used in previous studies), and by employing the IPMA within PLS-SEM, we clearly emphasized those particular aspects that are under the control of outdoor adventure parks’ managers, which significantly impact their visitors’ loyalty, as well as the way in which managers can clearly identify those attributes that need improvements.

Evidently, the current research has certain limitations. Firstly, the size of the sample included in our analyses is not generous. Nevertheless, when all the respondents are recent visitors of a single outdoor adventure park, and when the data collection follows an ad hoc procedure, large samples are hardly realistic. Secondly, our data were collected in 2019, before the COVID-19 pandemic, which most probably affected people’s attitudes toward traveling, as well as toward leisure activities in general. The effect of such attitude changes is obviously not reflected in our research. Thirdly, the generalizability of our results is rather limited to the context of outdoor adventure parks.

Each of our research limitations involves a suggestion for future research. Subsequent studies should process visitor data from several outdoor adventure parks, from various geographical regions. Future studies should also consider taking into account the effects of pandemic-based attitude changes regarding traveling and leisure activities, in the context of outdoor adventure parks. Additionally, future research could be conducted comparatively, using visitor data from several types of leisure parks (e.g., theme parks, national parks, adventure parks, etc.), in order to point out potential significant differences, and to enable researchers to provide more generalizable results.

## Figures and Tables

**Figure 1 ijerph-18-10033-f001:**
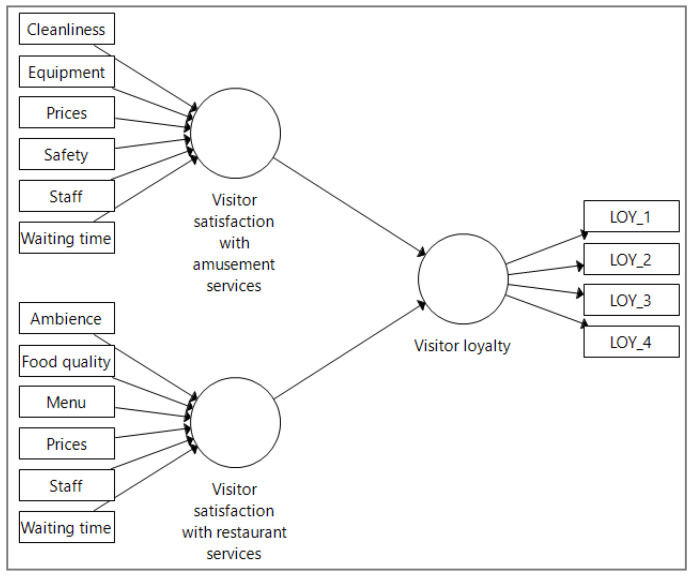
Proposed path model for the impact of visitor satisfaction on visitor loyalty.

**Figure 2 ijerph-18-10033-f002:**
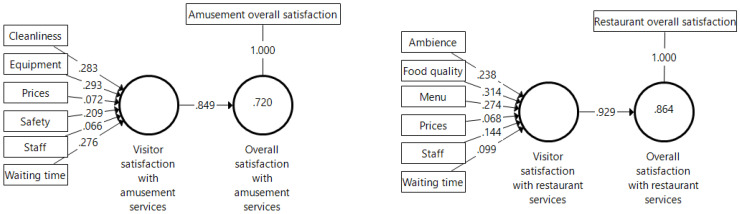
Redundancy analysis for assessing the convergent validity of formative variables. Note: Outer weights and path coefficients on arrows, R^2^ values in circles.

**Figure 3 ijerph-18-10033-f003:**
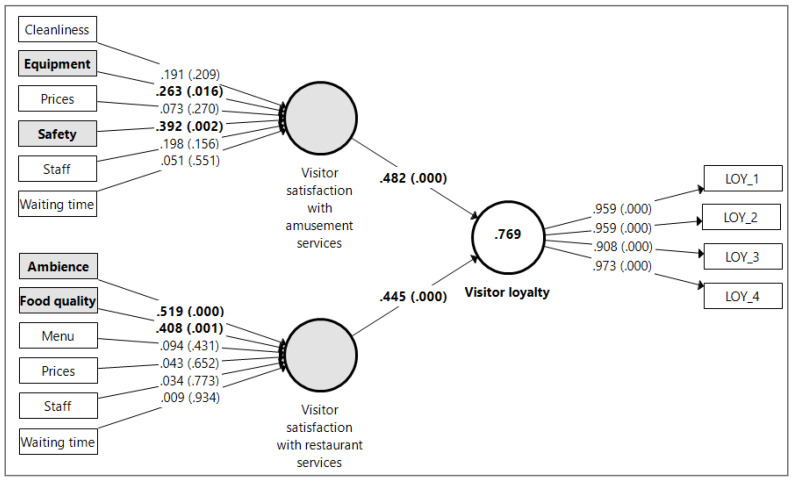
Structural model relationships assessment. Note: Results obtained after running the PLS-SEM bootstrapping procedure with 5000 subsamples. Values on arrows = outer weights (for visitor satisfaction), outer loadings (for the visitor loyalty), path coefficients (between constructs), *p* values (in parentheses). Value in circle = R^2^. Bold values = statistically significant.

**Figure 4 ijerph-18-10033-f004:**
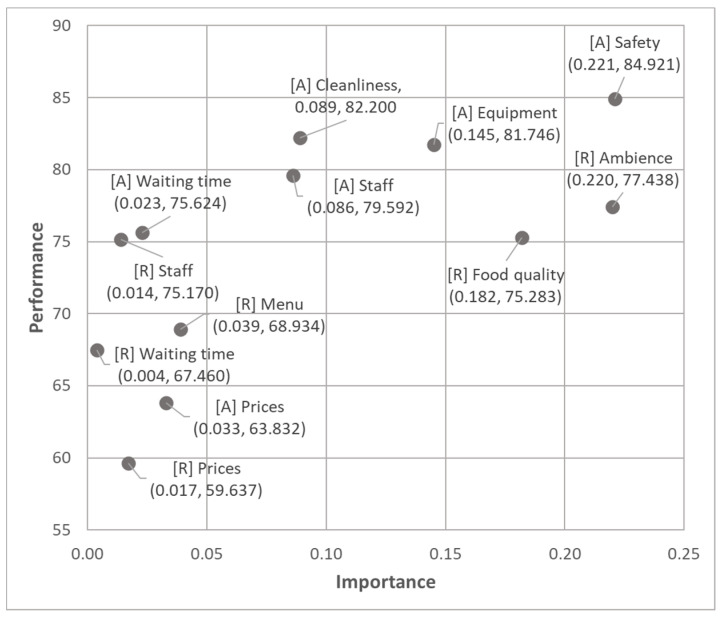
Importance–performance map analysis (IPMA) at indicator level. Note: Importance = unstandardized total effects; Performance = rescaled variables’ scores; [A] = Amusement services; [R] = Restaurant.

**Table 1 ijerph-18-10033-t001:** Previous research exploring the drivers of leisure parks’ visitors’ loyalty.

Study	Journal	Investigated Drivers of Visitor Loyalty	Park	Sample Size
[13]	Journal of Destination Marketing and Management	(1) physical environment; (2) interaction with staff; (3) interaction with other customers	Kuala Lumpur and Selangor, Malaysia	292
[14]	Tourism and Hospitality Research	(1) experiential quality (surprise, fun, immersion); (2) serviscape (communicative staging, substantive staging); (3) overall perceived service quality	Vialand, Istanbul	301
[15]	Tourism Management	(1) positive arousal; (2) positive disconfirmation; (3) pleasure	A Mediterranean theme park (NS)	200
[16]	International Journal of Tourism Research	(1) access quality; (2) technical quality; (3) physical environment; (4) personal interaction	Kinmen, Taiwan	616
[17]	Current Issues in Tourism	(1) quality of attractions; (2) quality of facilities; (3) staff service quality; (4) historical and cultural quality	Hangzhou Songcheng, China	314
[18]	Tourism Management	(1) serviscape (communicative staging, substantive staging)	Disneyland and Ocean Park, Hong Kong	366
[19]	Journal of Vacation Marketing	(1) overall park experience and value; (2) park food quality, value and variety; (3) park cleanliness, and atmosphere	A major US theme park (NS)	44,995
[20]	Tourism Review International	(1) “hygiene” factors; (2) quality of experience	Fjord&Bælt, Denmark	335
[21]	International Journal of Tourism Research	(1) experiential quality (immersion, surprise, participation, fun); (2) perceived value	A theme park in South Korea (NS)	376
[22]	Asia Pacific Journal of Tourism Research	(1) experiential quality (immersion, surprise, participation, fun); (2) theatrical elements (attractiveness, charm, performance, consistency)	Hualien Park, Taiwan	408
[23]	Social Behavior and Personality	(1) outcome fairness; (2) interactive fairness (following a service complaint)	Janfusun Fancyworld, Taiwan	208
[24]	Journal of Travel and Tourism Marketing	(1) appetitive goal congruence; (2) unexpectedness; (3) goal relevance; (4) goal interest (emotional dimensions)	Happy Valley, China	645
[25]	Journal of Destination Marketing and Management	(1) emotional stimulation (sense, feel, think, relate, act, flow); (2) value for money	Various US theme parks (NS)	371
[26]	Journal of Vacation Marketing	(1) intellectual needs; (2) relaxation needs; (3) social needs; (4) thrill rides; (5) novelty	Janfusan Fancyworld, Taiwan	402
[27]	Managing Service Quality	(1) ambiance; (2) interaction; (3) design; (4) joy	A theme park in eastern Norway (NS)	162
[28]	Journal of Hospitality and Tourism Research	(1) physical environment; (2) interaction quality; (3) outcome quality; (4) access quality	Janfusan Fancyworld, Taiwan	424

Note: NS = Not specified.

**Table 2 ijerph-18-10033-t002:** Sample demographics.

Gender		Age		Education	
Men	40.82%	<25 years	25.17%	Secondary	38.78%
Women	59.18%	25–34 years	31.97%	Higher	61.22%
		35–44 years	30.61%		
		>44 years	12.25%		
**Visit duration**			**Visitor residence**		
Day trip		57.14%	Hunedoara *		68.03%
Overnight		42.86%	Other Romanian counties		31.97%

* Hunedoara is the Romanian county in which Arsenal Park is located.

**Table 3 ijerph-18-10033-t003:** Collinearity assessment among the formative indicators.

Latent Formative Variables	Formative Indicators	Variance Inflation Factor (VIF)
Amusement services satisfaction	Cleanliness	2.945
	Equipment	2.343
	Prices	2.291
	Safety	3.026
	Staff	2.658
	Waiting time	1.833
Restaurant satisfaction	Ambience	2.556
	Food quality	4.526
	Menu	3.165
	Prices	2.314
	Staff	3.636
	Waiting time	3.221

**Table 4 ijerph-18-10033-t004:** Structural model predictive power assessment.

	Q^2^_predict _PLS_	RMSE _PLS_	RMSE _LM_	RMSE _PLS_ < RMSE _LM_
LOY_1	0.646	0.911	0.927	Yes
LOY_2	0.751	0.707	0.718	Yes
LOY_3	0.465	1.199	1.214	Yes
LOY_4	0.744	0.722	0.717	No

Note: PLSPredict procedure with 10 folds and 10 repetitions; RMSE _PLS_ = root mean squared error of prediction using PLS-SEM; RMSE _LM_ = root mean squared error of prediction using a naïve linear model.

**Table 5 ijerph-18-10033-t005:** Heterogeneity assessment.

Information Criteria	Number of Segments			
	1	2	3	4
AIC3	211.013	212.333	215.578	214.42
AIC4	214.013	219.333	226.578	229.42
BIC	216.985	226.266	237.473	244.276
CAIC	219.985	233.266	248.473	259.276
Segment	Segment’s relative size			
1	1	0.508	0.569	0.672
2		0.492	0.313	0.127
3			0.119	0.112
4				0.089

Note: FIMIX-PLS results using a stop criterion of 1 × 10^−5^, a maximum number of 5000 iterations, and 10 repetitions; AIC3(4) = Akaike’s Information Criterion Modified with Factor 3(4); BIC = Bayesian Information Criteria; CAIC = Consistent AIC.

## Data Availability

The data that support the findings of this study are available from the corresponding author upon reasonable request.

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
