# Peer review of "Exploring the Drivers of Visitor Loyalty in the Context of Outdoor Adventure Parks: The Case of Arsenal Park in Romania"

_ijerph, 2021, doi:10.3390/ijerph181910033_

Round 1

Reviewer 1 Report

The goal of this research is to explore those outdoor adventure park factors that affect the loyalty of tourists. The results show that safety, equipment, and catering are the most important factors. And through IMPA, clearly identify the needs and quality factors. However, it is recommended to modify the content to improve readability and research contribution. 1. The number of content items of the survey items in the questionnaire should be increased to understand the dimensions of the questionnaire. 2. Although Table 2 has the basic information of the survey respondents and understands the distribution of age and education level, whether transportation mode and income will also cause influencing factors, and whether it can be supplemented. 3. As a result of IMPA analysis, it is possible to know the importance of all interviewees for their needs and quality factors. In addition, the analysis of gender groups can already understand whether different ethnic groups have different rankings of needs and quality factors for IMPA. 4. The article explains the size and facilities of the outdoor adventure park. It should also be added to explain the mode of transportation and the price of tickets, so as to better understand the target audience of the park and the purpose of the park. 5. Figure 4 should explain how the performance value is calculated to understand the meaning of the value. And explain and explain the matrix in detail to understand the distribution of service functions.

Author Response

Dear Reviewer, 

First of all, we appreciate the time and effort you dedicated to providing feedback on our manuscript. We are truly grateful for your insightful comments, as well as for your improvement suggestions, which we found quite helpful. We have incorporated all your recommendations in our revised paper. Consequently, you will find various changes in the manuscript.

Thank you for your very professional recommendations. We are confident that the revised version of our paper is considerably improved, fully complying with your suggestions, with a much better flow, coherence, and narrative.

In the revised manuscript, we’ve emphasized in red all significant additions to and alterations of the initial version.

See attached, point-by-point, our responses to each of your comments and concerns.

Reviewer 2 Report

The structure of the manuscript is clear and easy to follow. The methodology provides specific details of assessment models and values, which is very helpful. The discussion/conclusion are also clear, linking to previous studies and discussing practical implications, limitations, and future research needs.

The only recommendation is to add a locator map of Arsenal Park and probably a park map to help contextualize readers.

Author Response

Dear Reviewer,

We appreciate the time and effort you dedicated to providing feedback on our manuscript. We have incorporated your recommendations in our revised paper. In the revised manuscript, we’ve emphasized in red all significant additions to and alterations of the initial version. 

See attached our response to your comments.

Reviewer 3 Report

The case-study is interesting and relevant as theme parks remain an under-researched domain.  Although the sample is relatively small for a PLS-SEM Study, the results are relevant and presented clearly.  Yet, there is plently of room for improvement here:

Research Rigour:

It is not transparent how the analysis of reviews (in conjunction with the literature review) resulted to a questionnaire (which is not included).  An additional issue is th representativeness of the sample.  Does it include only Romanian visitors or both Romanian and international?  The park also offers overnight accommodation.  Here, it could make a difference in terms of the factors examined whether the repsondents were day-trip visitors or overnight-visitors (also in terms of Food and Beverage).  

Research Relevance:  

Related to the sampling limitations and the methodological aspects measured, is the issue of implications for theory and practice.  The model tested is rather generic and offers little in terms of theoretical novelty.  That Safety and Food are key determinants of satisfaction in tourism, is hardly a novel finding.  The authors also highlight the cases, where the findings of the case study do not align with existing literature, whithou intenpreting or attempting to explain the differing results.   At this level of analysis, this study represents conventional satisfaction survey data, passed through a PLS-SEM software software; this is not necessarily problematic, but it needs to offer somekind of theoreical insight and contribution of this case study to the overall body of domain knowledge.

To summarise, this is a statistically sound study, which requires more transparency and elaboration in terms of model-building and a ritical discussion of results and their implications (not just description).

Author Response

(The authors gave the same response as above.)

Round 2

Reviewer 3 Report

The authors addressed the concerns raised in my original review.  A final issue, having an insight on the data collection and discussion, is the title of the paper, which is somewhat misleading.  The title is:  "Exploring the Drivers of Visitor Loyalty as Sustainability Requirement in the Context of Outdoor Adventure Parks".  The 'Sustainability requirement' is used as a euphimism for 'economic survival / success' while the paper does not adequately establish a connection between loyalty (or rather intention to re-visit) and the various other diemensions of sustainability (e.g. ecological, socio-cultural).  Plus, this refers to a single case study.  The title promises / implies a mich wider context / scope than the actual paper.  My suggestion would be: ""Exploring the Drivers of Visitor Loyalty in the Context of Outdoor Adventure Parks:  The Case of Arsenal Park in Hunedoara, Romania" 

Author Response

Dear Reviewer,

We have adjusted the title of our paper according to you suggestion (you’ll see the new title in red in the revised manuscript).

Indeed, it better reflects the actual content of the paper.

Thanks again for taking the time to carefully analyze our paper.